# Pharmacophore hybridisation and nanoscale assembly to discover self-delivering lysosomotropic new-chemical entities for cancer therapy

Zhao Ma [1,2], Jin Li[1], Kai Lin [1], Mythili Ramachandran[1], Dalin Zhang[1], Megan Showalter[3], Cristabelle De Souza[1], Aaron Lindstrom[1], Lucas N. Solano[1], Bei Jia[1], Shiro Urayama [4], Yuyou Duan[5], Oliver Fiehn [3], Tzu-yin Lin[6], Minyong Li [2,7] & Yuanpei Li [1✉]

Integration of the unique advantages of the fields of drug discovery and drug delivery is invaluable for the advancement of drug development. Here we propose a self-delivering one-component new-chemical-entity nanomedicine (ONN) strategy to improve cancer therapy through incorporation of the self-assembly principle into drug design. A lysosomo-tropic detergent (MSDH) and an autophagy inhibitor (Lys05) are hybridised to develop bisaminoquinoline derivatives that can intrinsically form nanoassemblies. The selected BAQ12 and BAQ13 ONNs are highly effective in inducing lysosomal disruption, lysosomal dysfunction and autophagy blockade and exhibit 30-fold higher antiproliferative activity than hydroxychloroquine used in clinical trials. These single-drug nanoparticles demonstrate excellent pharmacokinetic and toxicological profiles and dramatic antitumour efficacy in vivo. In addition, they are able to encapsulate and deliver additional drugs to tumour sites and are thus promising agents for autophagy inhibition-based combination therapy. Given their transdisciplinary advantages, these BAQ ONNs have enormous potential to improve cancer therapy.

[1] Department of Biochemistry and Molecular Medicine, UC Davis Comprehensive Cancer Center, University of California Davis, Sacramento, CA 95817, USA. [2] Department of Medicinal Chemistry, Key Laboratory of Chemical Biology (MOE), School of Pharmacy, Cheeloo College of Medicine, Shandong University, Jinan 250012 Shandong, China. [3] West Coast Metabolomics Center, University of California Davis, Davis, CA 95616, USA. [4] Division of Gastroenterology and Hepatology, Department of Internal Medicine, University of California Davis, Sacramento, CA 95817, USA. [5] Institutes for Life Sciences, School of Medicine, South China University of Technology, Guangzhou 510006 Guangdong, China. [6] Division of Hematology and Oncology, Department of Internal Medicine, University of California Davis, Sacramento, CA 95817, USA. [7] State Key Laboratory of Microbial Technology, Shandong University, Jinan 250100 Shandong, China. ✉email: lypli@ucdavis.edu

The growing exploration of nanomedicine has contributed greatly to cancer treatment over the past few decades[1]. Both carrier-assisted nanomedicines and carrier-free nanomedicines are being developed to improve drug-intrinsic kinetics and safety profiles[2–5]. However, there are several limitations associated with these nanotherapeutic approaches[6,7]. First, complexity and toxicity due to their multicomponent natures have severely hampered the clinical translation of many nanoformulations[6]. Second, most drugs used in conventional delivery studies were approved decades ago and some are no longer first-line treatments[5,8]. Third, nanoformulations that are designed for recently developed therapeutic agents, especially new-chemical entities, may encounter patent issues. Finally, not all drugs can be structurally modified. These limitations can be addressed in the context of medicinal chemistry[9]. Compared with nanomedicine, which focuses on delivery profiles for drug research and development, medicinal chemistry commits to the discovery of drug entities in earlier stages[10]. Although drug discovery technologies have generated numerous drug leads and candidates, problems surrounding drug kinetics, metabolism and toxicology remain challenging[11,12]. These challenges may also be solved relatively easily by nanotechnologies from the field of nanomedicine. To take advantage of this transdisciplinary connection, we herein integrate the principle of nanotechnology into initial drug design and develop a one-component new-chemical-entity nanomedicine (ONN) strategy (Fig. 1a). In this strategy, the drug design follows both conventional drug design strategies and molecular self-assembly principles so that designed drugs are endowed with advantages from the perspectives of both drug discovery and drug delivery.

To perform a proof-of-concept experiment, we chose lysosomes as therapeutic cancer targets[13–15]. Cancer cell lysosomes are hypertrophic and easily ruptured and are more fragile than normal lysosomes[16]. Lysosomal membrane permeabilization (LMP) can directly trigger cell death by enabling the release of proteolytic enzymes (i.e. cathepsins) into the cytoplasm; therefore, lysosomotropic detergents that can induce LMP have been developed for tumour treatment[17]. Moreover, lysosomal inhibition has considerable potential as an anticancer strategy because it interferes with autophagy, an important pathway for the stress response and drug resistance of established tumours[18–22]. The lysosomotropic alkalizers chloroquine (CQ) and hydroxychloroquine (HCQ) are commonly used autophagy inhibitors that have been tested in multiple clinical trials against various cancer types. However, their efficacy is considered insufficient, particularly when they are used as single agents[23]. To discover more effective autophagy inhibitors, McAfee et al. synthesised a bisaminoquinoline (BAQ) derivative, Lys05, that can impair tumour growth both in vitro and in vivo as a single agent[24].

In this work, we adopt the strategies of pharmacophore hybridisation and molecular self-assembly to design a series of lipophilic cationic BAQ derivatives (Fig. 1b and Supplementary Fig. 1)[25,26]. BAQ12 and BAQ13 are selected to construct ONNs because of their potential to be therapeutic agents and self-assembling building blocks. These BAQ ONNs display excellent anticancer activity in vitro, with enhanced effects on lysosomal disruption, lysosomal dysfunction and autophagy inhibition. Moreover, as nanodrugs, the BAQ ONNs exhibit the expected self-delivering profiles. These advantages from the perspectives of both drug discovery and drug delivery ultimately contribute to

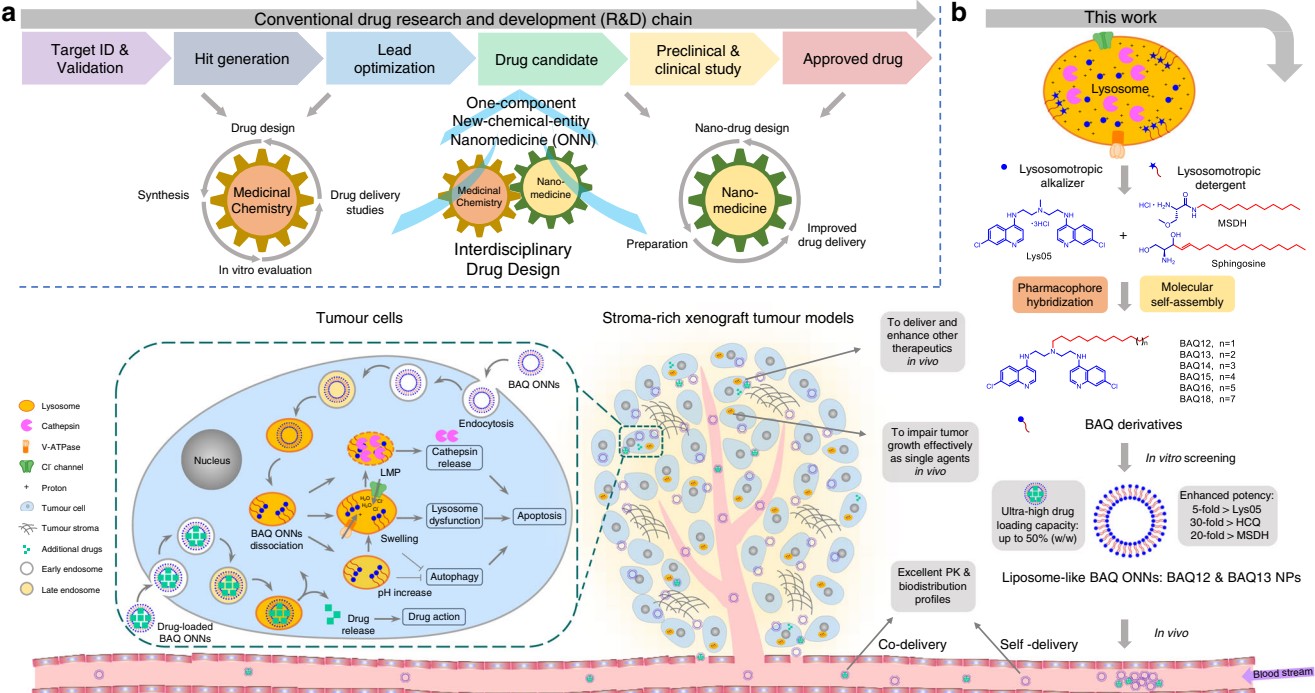

**Fig. 1 Schematic illustration of the proposed drug design strategy and the current work. a** An interdisciplinary drug design strategy is proposed to integrate the conventional fields of medicinal chemistry and nanomedicine. Drugs are named as one-component new-chemical-entity nanomedicines (ONNs), which are designed according to the strategies of conventional drug design and molecular self-assembly so that they could acquire the advantages from the perspectives of both drug discovery and drug delivery. **b** The proof-of-concept experiment in this work: discovery of self-delivering lysosomotropic bisaminoquinoline (BAQ) derivatives for cancer therapy. The BAQ derivatives, generated from the hybridisation of lysosomotropic detergents and the BAQ-based autophagy inhibitor, can self-assemble into BAQ ONNs that show enhanced functions in vitro, excellent delivery profiles and significant in vivo therapeutic effects as single agents. Moreover, they also possess high drug-loading efficiency to deliver the additional drug into tumour sites, thus generating a promising application of combination therapy.

the significant anticancer activity of these compounds as single agents in gastrointestinal cancer models in vivo. In addition, the BAQ ONNs display promise for applications in combination therapy with napabucasin, as they play dual roles as both therapeutic agents and delivery carriers. With their multidisciplinary integration and ingenious functional superposition, BAQ ONNs will emerge as good alternatives for improvement of cancer treatment.

## Results

**Discovery of BAQ derivatives as potential ONNs.** BAQ12–BAQ18 were designed via hybridisation of the key structural elements of the lysosomotropic autophagy inhibitor Lys05 and the lysosomotropic detergent MSDH to achieve pharmacological fusion (Fig. 1b). Based on self-assembly principles, we envisioned that the inclusion of long hydrophobic tails with the cationic BAQ heads would drive them to form nanoparticles (NPs)[26,27]. Since BAQ heads have a calculated pKa of 8.4, this self-assembly should be dependent on the surroundings' pH, wherein NPs are formed under neutral conditions and are dissociated into free building blocks after protonation in acidic environments.

The compounds (BAQ12–BAQ18) were synthesised and structurally confirmed by $^1H$ NMR, $^{13}C$ NMR and HRMS spectra (Supplementary Fig. 1). In contrast to Lys05 and MSDH, BAQ12–BAQ18 were not completely soluble in water as free base or hydrochloride salt forms, which is a definite defect that prevents them from being ideal drugs. However, the lipophilic cations allowed for spontaneous self-assembly in water via nanoprecipitation, which resulted in homogeneous opalescent NP solutions. The assembled NPs of BAQ12–BAQ18 had similar nanoscale characteristics, including their sizes (100–140 nm), polydispersity index (PDI) values <0.1, and positive surface charges (~+40 mV) (Table 1 and Supplementary Fig. 2a). The pH-responsive dissociation behaviour was then assessed by monitoring the size changes of the particles (Fig. 2a). All BAQ NPs were intact under near-neutral conditions and dissociated under only relatively acidic conditions. The critical dissociation pH was 5.5–6.0 for BAQ12–BAQ14 and 5.0–5.5 for BAQ15–BAQ18. When protonated in an acidic environment, the BAQ12–BAQ18 lipophilic cations turned into amphiphilic molecules and acquired surface activity. A haemolysis test was then utilised to evaluate the pH-responsive biomembrane disruption ability of the compounds[28]. None of the NPs had haemolytic effects under approximately neutral conditions (pH ≥ 6.5) but started to induce haemolysis when the pH was <6.0 (Fig. 2b and Supplementary Fig. 2b). Among the compounds, BAQ12 NPs and BAQ13 NPs exhibited the strongest haemolytic activity, inducing up to 90% haemolysis under simulated lysosomal conditions (pH 4.0–5.5); in contrast, BAQ14 NPs induced moderate

haemolysis (70%), and BAQ15–BAQ18 NPs only yielded ~50% haemolysis. In the control groups, the conventional lysosomal detergent MSDH exhibited only a weak haemolytic response to pH, and Lys05 without detergence did not elicit observable haemolysis in the whole pH range at the same concentration. Because LMP is a potential stimulus for apoptosis, BAQ12 and BAQ13, the detergence of which can be activated in lysosomes, might be effective in inducing cancer cell death directly. Furthermore, upon titration with hydrochloride (HCl), BAQ12 NPs and BAQ13 NPs displayed obvious pH plateaus within a narrow pH range (at ~pH 6.0), indicating their strong pH buffering capacity (Fig. 2c)[29]. In contrast, the pH values of the other NPs (BAQ14–BAQ18) decreased proportionally, and only short pH plateaus were observed for Lys05 (pH 7.2) and MSDH (pH 6.2). Since sufficient acidification is required for lysosomal degradation, BAQ12 and BAQ13, with their strong $H^+$ buffering capacity, showed greater potential than the other compounds to induce lysosomal dysfunction and could therefore impair tumour cell growth.

To verify the therapeutic effects of BAQ12–BAQ18, we conducted a preliminary screening using an MTS assay on various cancer cell lines. Within 24 h treatment, these derivatives exhibited anti-proliferative effects at different levels. BAQ12 and BAQ13 were highly effective and showed ~3-fold, ~20-fold and ~10-fold higher potency than Lys05, HCQ and MSDH, respectively, but the activity decreased steadily as the hydrophobic tails extended from 14 to 18 carbons (Table 1 and Supplementary Fig. 2c). This decrease was due to gradual declines in the detergence and $H^+$ buffering capacity of compounds. Based on the results above, BAQ12 and BAQ13 were then selected as representatives for construction of BAQ ONNs in the following studies.

**pH-responsive assembly and high drug-loading efficiency.** The pH-responsive assembly dissociation phase transition of BAQ ONNs was then determined by transmission electron microscopy (TEM). At pH 7.4, the NPs exhibited a strong Tyndall effect and displayed liposome-like nanostructures with ~100 nm diameters and bilayer thicknesses of ~5 nm (Fig. 2d). These results were consistent with the dynamic light scattering measurements. In contrast, at pH 5.0, the solution lost its Tyndall effect, and the vesicles were absent under TEM, which demonstrated that the NPs were dissociated under this condition (Fig. 2e). The release behaviour of BAQ ONNs at physiological pH (7.4) and lysosomal pH (5.0) was then investigated. As shown in Fig. 2f, BAQ12 NPs and BAQ13 NPs were released almost completely over 8 h (~90%) at pH 5.0, but under the neutral condition, only ~10% agents were released over 24 h. Considering that lysosomes maintain a pH in the range of 4.0–5.5, we believe that BAQ

**Table 1 Nanoparticle characterisation and IC$_{50}$ values on cancer cells of BAQ derivatives.**

| Compound | Size (nm) | PDI | Zeta potential (mV) | IC$_{50}$ (μM) in the 24 h treatment | | | | | |
|---|---|---|---|---|---|---|---|---|---|
| | | | | MIA PaCa-2 | PANC-1 | BXPC-3 | HT29 | H460 | MCF-7 |
| BAQ12 | 102.5 ± 3.1 | 0.069 | 36.4 ± 1.5 | 4.1 ± 0.8 | 4.5 ± 0.9 | 2.9 ± 0.6 | 3.5 ± 1.2 | 2.7 ± 0.5 | 3.1 ± 0.1 |
| BAQ13 | 99.1 ± 2.1 | 0.097 | 39.7 ± 1.9 | 4.2 ± 1.0 | 4.3 ± 1.0 | 3.1 ± 0.2 | 3.0 ± 1.1 | 2.2 ± 0.4 | 3.7 ± 0.3 |
| BAQ14 | 137.9 ± 4.6 | 0.069 | 39.1 ± 0.4 | 16.0 ± 1.8 | 10.4 ± 1.8 | 7.5 ± 0.7 | 9.6 ± 2.0 | 24.0 ± 0.3 | 28.5 ± 3.5 |
| BAQ15 | 107.3 ± 2.6 | 0.086 | 38.0 ± 2.9 | 42.0 ± 11.7 | 34.5 ± 3.9 | 17.2 ± 2.0 | 16.1 ± 2.5 | 24.1 ± 0.4 | >60 |
| BAQ16 | 119.7 ± 2.1 | 0.082 | 40.4 ± 0.5 | 60.7 ± 13.9 | >90 | 27.6 ± 2.8 | >80 | >90 | >90 |
| BAQ18 | 96.7 ± 1.6 | 0.093 | 39.5 ± 0.8 | >90 | >90 | >90 | >100 | >90 | >90 |
| Lys05 | – | – | – | 16.7 ± 4.7 | 13.3 ± 2.0 | 7.8 ± 0.9 | 10.6 ± 3.4 | 11.1 ± 0.6 | 12.3 ± 0.7 |
| HCQ | – | – | – | 79.6 ± 8.5 | – | – | >75 | – | – |
| MSDH | – | – | – | 55.5 ± 7.1 | – | – | 30.3 ± 4.8 | – | – |

(Data are mean values ± SD, $n$ = 3.)

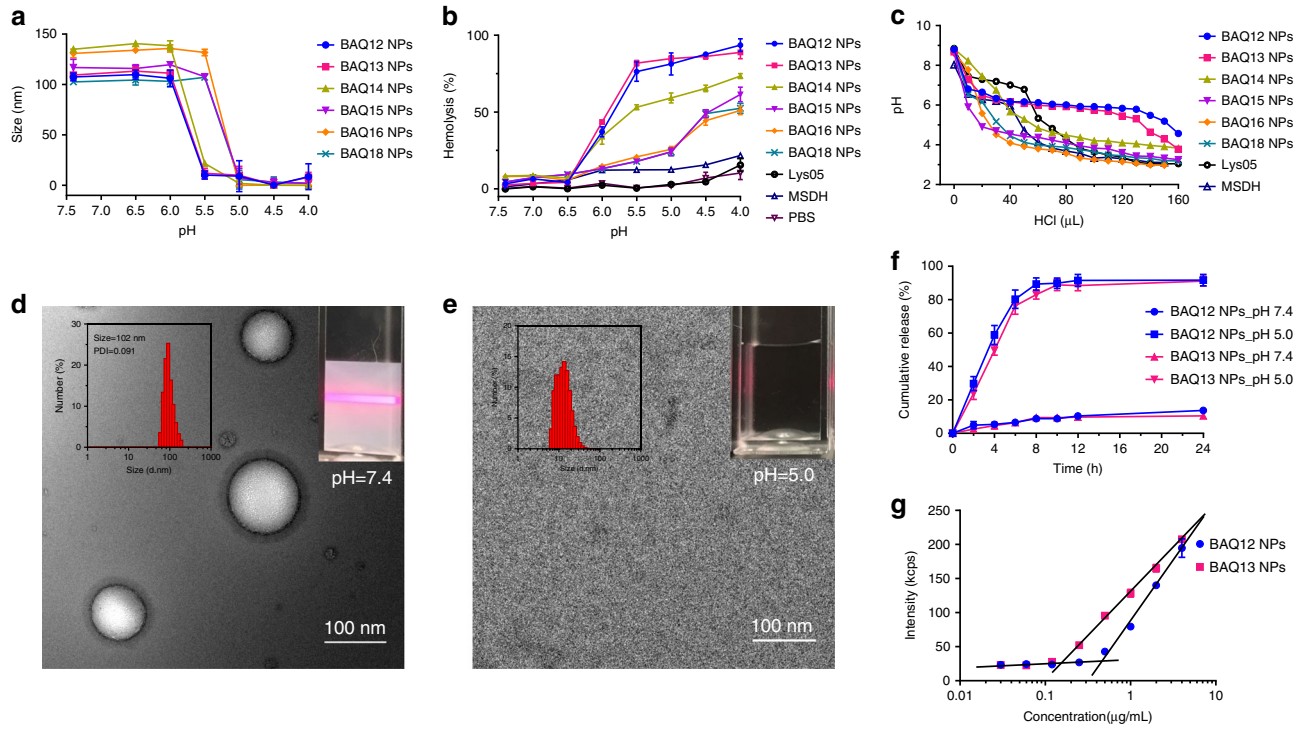

**Fig. 2 Characterisation of BAQ ONNs. a** Size change of BAQ NPs (10 μM) in acetate buffer with different pH values; data are mean values ± SD; $n = 3$ independent nanoparticle samples. **b** The pH-dependent haemolysis induced by BAQ NPs (50 μM, 4 h) in PBS buffer; data are mean values ± SD; $n = 3$ independent nanoparticle samples. **c** The pH change of BAQ NPs (1 mM) within hydrochloric acid (HCl, 0.1 M) titration. **d**, **e** Representative TEM micrograph at pH 7.4 (**d**) and pH 5.0 (**e**); the insets display the size distribution (left) and Tyndall effect (right); experiments were all repeated three times independently. **f** In vitro drug releasing patterns at pH 7.4 and pH 5.0; data are mean values ± SD; $n = 3$ independent nanoparticle samples. **g** The count rate for various concentrations of BAQ NPs in water; The intersection of two lines refers to CACs of BAQ12 NPs (0.45 μg mL$^{-1}$, 0.76 μM) and BAQ13 NPs (0.15 μg mL$^{-1}$, 0.25 μM); data are mean values ± SD; $n = 3$ independent nanoparticle samples.

**Table 2 Parameters of drug loading using BAQ13 NPs.**

| Drugs or dyes | NPs : drug (mass ratio) | Drug-loading content | Encapsulation efficiency | Size/diameter | PDI |
|---|---|---|---|---|---|
| DiD (dye) | 1 : 1 | 50% | 100% | 120 nm | 0.1 |
| Bortezomib | 1 : 1 | 50% | 89% | 92 nm | 0.09 |
| β-lapachone | 2 : 1 | 33% | 95% | 128 nm | 0.047 |
| JQ1 | 1 : 1 | 50% | 92% | 110 nm | 0.048 |
| Rapamycin | 1 : 1 | 50% | 90% | 85 nm | 0.15 |
| Etoposide | 1 : 1 | 50% | 86% | 69 nm | 0.104 |
| Apoptozole | 1 : 1 | 50% | 93% | 85 nm | 0.099 |
| Vinblastine | 1 : 1 | 50% | 95% | 60 nm | 0.133 |
| Lenalidomide | 1 : 1 | 50% | 88% | 85 nm | 0.069 |
| Napabucasin | 4 : 1 | 20% | 85% | 110 nm | 0.101 |

ONNs will dissociate into free small molecules upon arrival in these compartments and will thus exert therapeutic effects. The critical aggregation concentrations (CACs) of BAQ12 NPs and BAQ13 NPs were measured to be 0.76 and 0.25 μM, respectively (Fig. 2g)[30]. The threefold difference observed between them indicated that BAQ13 could form NPs more easily than BAQ12 despite a difference of only one methylene unit between their molecular structures. The two NPs also exhibited decent stability in particle size over a relatively long duration at room temperature, even in the presence of 10% serum or 0.5 mM bovine serum albumin (Supplementary Fig. 3a–f). In addition, BAQ13 NPs showed higher stability than BAQ12 NPs in such long-term storage, which is likely due to their different CACs.

We next sought to investigate whether liposome-like BAQ ONNs can encapsulate additional agents. Upon nanoprecipitation

of BAQ13 and various agents, homogeneous NPs with mono-modal size distributions spontaneously formed (Table 2 and Supplementary Fig. 3g). BAQ13 NPs exhibited high drug-loading content (up to 50%, mass ratio) along with ~90% drug encapsulation efficiency (Table 2), which indicated that BAQ13 NPs could surpass the drug-loading limitations of the conventional liposome- and polymeric-based drug delivery systems[3]. It is very encouraging that these simple NPs composed of single small-molecule therapeutic entities exhibit such a powerful drug-loading capacity.

**Accumulation in lysosomes and lysosomal disruption.** To verify the lysosomal accumulation of BAQ ONNs, the near-infra-red fluorescent dye, 1,1′-dioctadecyl-3,3,3′,3′-tetramethylindodi-carbocyanine (DiD) was loaded for labelling and tracking[31]. As

expected, the lysosome puncta (green) in MIA PaCa-2 cells stained by Dextran-Alexa Fluor 488 (AF488) overlapped consistently with the DiD-labelled NPs (red), suggesting that BAQ ONNs were quickly taken up by cells and accumulated in lysosomes (Fig. 3a and Supplementary Fig. 4a). Upon this accumulation, BAQ ONNs reduced the LysoTracker-positive puncta, showing their ability to deacidify lysosomes similarly to Lys05 and MSDH (Fig. 3b and Supplementary Fig. 4b, c)[24].

The induction of LMP by BAQ12 and BAQ13 was investigated by live cell staining using the dye acridine orange (AO)[32]. Compared to those treated with Lys05 and MSDH, the cells treated with BAQ12 NP or BAQ13 NPs exhibited reduced numbers of red puncta and increased ratios of green to red fluorescence, suggesting that BAQ ONNs have an increased capability to induce lysosomal disruption in cancer cells. (Fig. 3c, and Supplementary Fig. 4d, e). This LMP effect was further confirmed by detecting the release of Dextran-AF488 from lysosomes[33]. As shown in Fig. 3d, treatment with BAQ ONNs resulted in a diffuse staining pattern throughout the cytoplasm, indicating lysosomal leakage, whereas the fluorescence in control cells appeared restricted to punctate structures, representing intact lysosomes. With their LMP function, BAQ ONNs were demonstrated to induce the release of cathepsin B from isolated lysosomes, which is an important trigger of apoptosis (Fig. 3e)[34]. As LMP was not observed in MSDH-treated cells, the results suggested that BAQ12 and BAQ13 represent the next generation of lysosomotropic detergents[17].

**Autophagy inhibition**. To explore the effect of BAQ ONNs on autophagy, we measured the levels of microtubule-associated protein 1 light chain 3 (LC3) and Sequestosome 1 (SQSTM1)/p62 protein, which are often used to monitor changes in the autophagy process[35]. During autophagy, the cytosolic form of LC3 (LC3-I) is converted into the lipid modified form (LC3-II), which is then recruited to the autophagosomal membrane. Meanwhile, the autophagy substrate SQSTM1/p62 protein is degraded via selective incorporation into autophagosomes. Therefore, increased levels of both LC3-II and SQSTM1/p62 should be observed when autophagy is inhibited, while increased LC3-II levels and decreased SQSTM1/p62 levels should be observed if autophagy is activated. As shown in Fig. 3f, g, compared to the untreated cells and Lys05-treated cells, MIA PaCa-2 cells treated with BAQ ONNs showed significant concentration-dependent increases in both LC3B-II and SQSTM1/p62 protein levels. Such increases were also observed after treatment with bafilomycin A1 (BfA1), a known autophagy inhibitor[36]. These findings indicate that BAQ ONNs can inhibit cellular autophagy more effectively than Lys05.

The autophagy-inhibiting effect was then confirmed by using LC3B-GFP imaging, as the formation of fluorescent LC3-II puncta in cells can be used to visualise the accumulation of autophagosomes[24]. The cells treated with BAQ ONNs generated conspicuous LC3B-GFP puncta in a concentration-dependent manner (Fig. 3h and Supplementary Fig. 4f). The LC3B-GFP puncta per cell were quantified, which revealed the higher autophagy-inhibition potency of BAQ ONNs than Lys05 (Fig. 3i). For further verification, we also used TEM to monitor cell micromorphological changes. As expected, compared to Lys05 and MSDH, BAQ ONNs induced the formation of larger autophagic vesicles (AVs) or autophagosomes in cells, which further confirmed the improved autophagic inhibition effects of BAQ ONNs (Fig. 3j, k). Taken together, the findings indicated that BAQ12 NPs and BAQ13 NPs surpassed the parental Lys05 in inhibiting autophagy; thus, BAQ ONNs represent a generation of nanoformulated autophagy inhibitors.

**Proton-sponging effect and lysosomal dysfunction**. As cationic molecules, both BAQ12 and BAQ13 possess strong $H^+$ buffering capacity, an essential characteristic of materials with proton-sponging effects (Fig. 2c)[37,38]. The TEM results above indicated that the BAQ ONNs could significantly enlarge lysosomes (Fig. 3j, k), demonstrating the proton-sponging effects of BAQ ONNs. To further investigate these effects, we characterised the transcriptomic changes in MIA PaCa-2 cells post treatment using RNA sequencing (RNA-seq). A total of 13234 genes were tested, and their expression levels were compared among vehicle, Lys05 and BAQ13 groups. Using volcano plot analysis, we found 165 differentially expressed genes (DEGs) (fold change $\geq 2$ and $p$ value $\leq 0.05$) in the Lys05 group compared with the vehicle group, including 62 upregulated genes and 103 downregulated genes. In comparison, 390 DEGs were found in the BAQ13-treated cells, including 209 upregulated genes and 181 downregulated genes (Supplementary Fig. 5a). Enrichment analysis of the gene set with the Kyoto Encyclopaedia of Genes and Genomes (KEGG) revealed that BAQ13 NPs induced robust upregulation of lysosome-associated genes, such as *V-ATPase*, *$Cl^-$ channel*, *protease*, and *lysosome-associated membrane protein (LAMP)* genes (Fig. 4a–c and Supplementary Fig. 5b)[38]. qPCR analysis also confirmed the upregulation of *V-ATPase* and *$Cl^-$ channel* genes, which remarkably indicated that the BAQ ONNs had strong proton-sponging properties (Fig. 4d, e).

The upregulation of important lysosomal enzyme genes, such as *cathepsin* and *NEU1*, also emphasized on the lysosomal dysfunction caused by BAQ ONNs (Fig. 4b, c). *LAMP* genes that are thought to be partly responsible for maintaining lysosomal integrity were upregulated as well, which indicated the function of BAQ ONNs in lysosomal disruption (Supplementary Fig. 5c). The BAQ ONN treatment groups exhibited high transcriptomic levels of proapoptotic genes (*BAX*, *BAK1*, *BAD*, *BIM* and *PUMA*), revealing the enhanced proapoptotic effects (Supplementary Fig. 5c, d). The BAQ ONN-induced lysosomal dysfunction was then confirmed using lipidomic analysis[39]. BAQ13 NPs induced accumulation of the acid sphingomyelinase (ASM) precursor sphingomyelin (SM) and led to decreases in the levels of its product, ceramide (Cer) (Supplementary Fig. 5e). In addition, the levels of phospholipase A (PLA) precursors, including phosphatidylcholine (PC), phosphatidylethanolamine (PE) and phosphatidylserine (PI), were decreased and the levels of their corresponding products, lysophosphatidylcholine (LPC), lyso-phosphatidylethanolamine (LPE) and lysophosphatidylserine (LPI), were increased (Supplementary Fig. 5f).

**In vitro antitumour activity**. To systematically investigate the antitumour effects in vitro, three pancreatic cancer cell lines (MIA PaCa-2, BxPC-3 and PANC-1) and two colon cancer cell lines (HT29 and HCT116) were selected for a 48 h MTS assay[24]. The BAQ ONNs showed $IC_{50}$ values of 1–3 μM and were thus ~5-fold, 30-fold and 20-fold more potent than Lys05, HCQ and MSDH, respectively (Fig. 4f, Supplementary Fig. 6, and Supplementary Table 1). These results also indicated that treatment with BAQ12 or BAQ13 alone was more effective than combination treatment with Lys05 and MSDH, suggesting that pronounced pharmacodynamic synergism occurred upon pharmacophore fusion. The results of cell growth and colony formation assays further demonstrated the inhibitory effects of BAQ ONNs on tumour cells (Fig. 4g, h). The improved anticancer activity of BAQ ONNs is attributable to their multiple functions in inducing LMP, lysosomal dysfunction and autophagy inhibition in cancer cells; these effects are considered to be important triggers of apoptosis. To examine the proapoptotic effects of BAQ ONNs, we subsequently detected apoptosis signals in MIA PaCa-2 and HT29 cells. BAQ ONN treatment resulted in significant

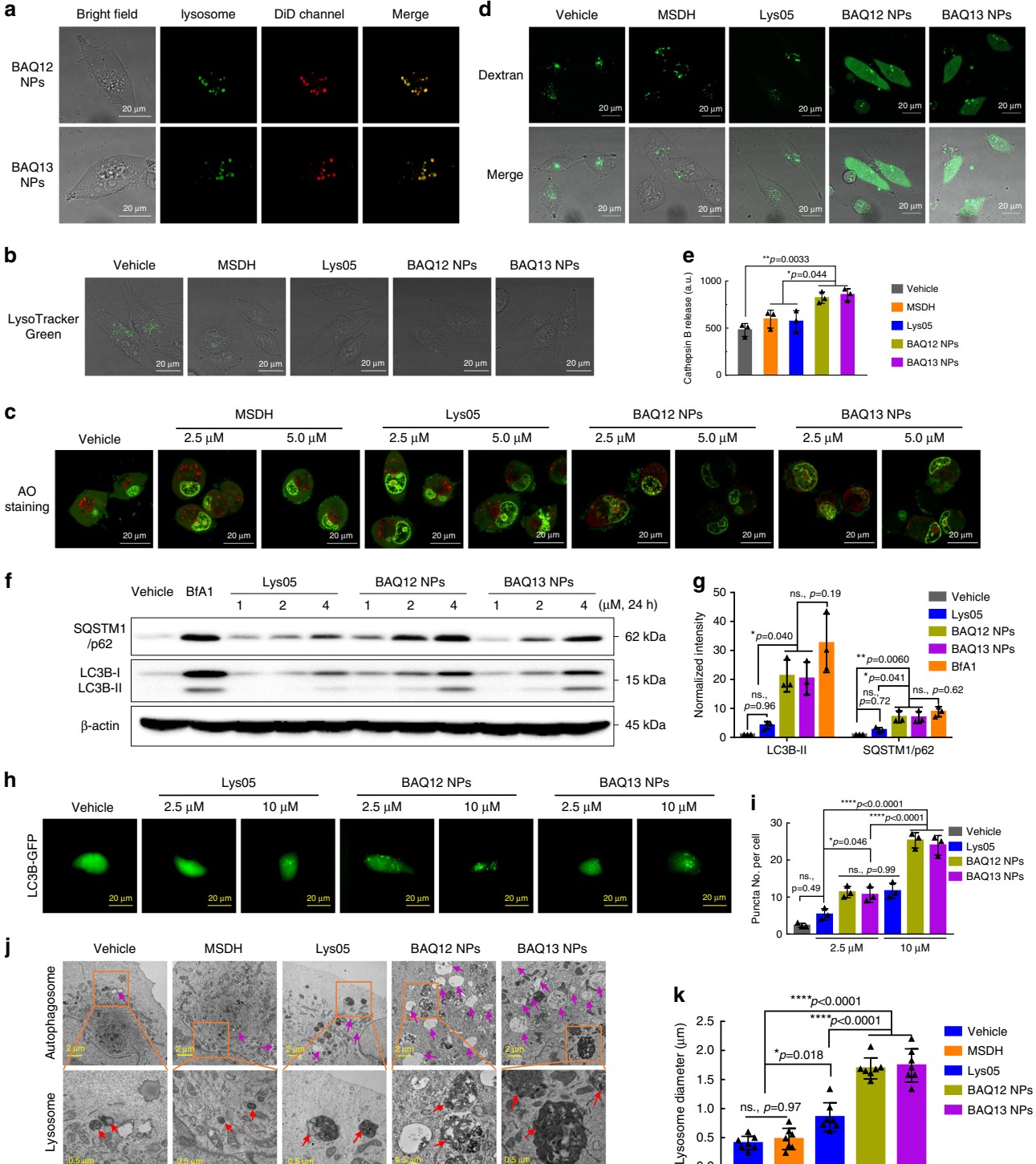

**Fig. 3 BAQ ONNs induced lysosomal disruption and inhibited autophagy in MIA PaCa-2 cells. a** Representative images for cellular uptake of nanoparticles; Dextran-AF488-loaded cells were incubated with DiD-labelled BAQ ONNs for 2 h; experiments were repeated three times independently. **b** Cells were treated as indicated (10 μM, 2 h) and were stained by LysoTracker Green; experiments were repeated three times independently. **c** AO staining of cells within the indicated treatments (5 μM, 12 h); experiments were repeated three times independently. **d** Representative images of Dextran-AF488-loaded cells that were treated as indicated (5 μM, 12 h); experiments were repeated three times independently. **e** Cathepsin B release from isolated lysosomes after treatments as indicated (25 μM, 12 h); data are presented as mean values ± SD; $n = 3$. **f** Western blotting. **g** Normalised quantification analysis of gel blots in **f**; data are presented as mean values ± SD; $n = 3$. **h** Representative LC3B-GFP images for the indicated 4 h treatments; experiments were repeated three times independently. **i** Quantification of LC3B-GFP puncta per cell in **h**; data are presented as mean values ± SD; $n = 3$. **j** Representative TEM images of cells that were treated as indicated (2 μM, 48 h); orange rectangle: region of interest; purple arrows: autophagic vesicles; red arrows: lysosomes. **k** The average diameter of lysosomes; data are presented as mean values ± SD; $n = 7$. All statistical $p$ values were calculated by one-way ANOVA with the Tukey's multiple comparison test. ns., not significant; *$p < 0.05$; **$p < 0.01$; ****$p < 0.0001$.

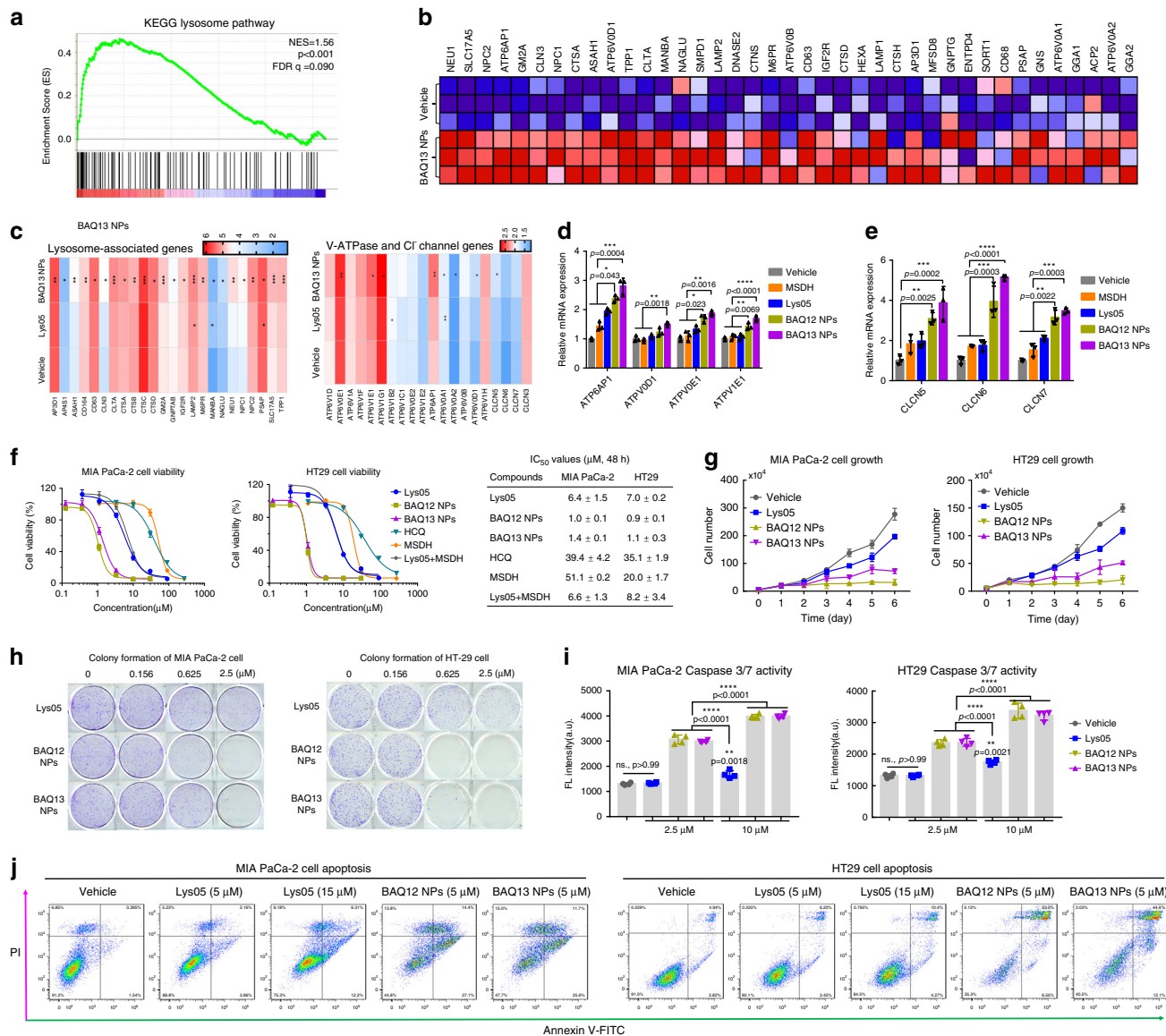

**Fig. 4 BAQ ONNs altered the expression of lysosomal genes and caused cell death via apoptosis. a** GSEA demonstrating the enrichment of lysosomal gene sets in MIA PaCa-2 cells treated with BAQ13 NPs (5 μM, 24 h). GSEA was performed with $n = 1000$ permutations, where $p$ adjust < 0.05 and FDR < 0.05 were considered significant. **b** Representative upregulated lysosomal genes from **a**. **c** Comparison of gene upregulation between Lys05 and BAQ13 NPs. **d**, **e** qPCR analysis of *V-ATPase* genes (**d**) and *Cl⁻ channel* genes (**e**) involved in the indicated treatments (5 μM, 24 h); data are mean values ± SD; $n = 3$. **f** Viability curves of cells that were exposed to the 48 h treatments and the corresponding IC$_{50}$ values; data are mean values ± SD; $n = 3$. **g** MIA PaCa-2 (1.5 μM) and HT29 (1.0 μM) cell growth curves within continuous treatments; data are mean values ± SD; $n = 3$ independent experiments. **h** Clonogenic assay of MIA PaCa-2 and HT29 cells; $n = 3$. **i** Caspase 3/7 activity in MIA PaCa-2 and HT29 cells that were treated for 6 and 12 h, respectively; Data are mean values ± SD; $n = 4$. **j** Percentage of apoptotic population of MIA PaCa-2 (left) and HT29 (right) cells that were treated for 24 h. All the statistical $p$ values were calculated by one-way ANOVA with the Tukey's multiple comparison test; ns not significant; *$p < 0.05$; **$p < 0.01$; ***$p < 0.001$; ****$p < 0.0001$.

elevations in both caspase 3/7 activity and apoptosis levels (Fig. 4i, j). Lys05, the control, increased apoptotic signals in a concentration-dependent manner, but its effect at a high concentration close to the IC$_{50}$ was still milder than those of the low concentrations of BAQ ONNs. These results demonstrate that cancer cells are more likely to undergo apoptosis when treated with multifunctional BAQ entities than when treated with Lys05, whose main function is autophagy inhibition. In addition, compared to the panel of cancer cell lines, the non-cancerous cell lines including IMR-90 cells, NIH/3T3 cells, and bone marrow cells, showed relative insensitivity to BAQ ONNs, thus indicating the relative high safety of those compounds (Supplementary Fig. 6 and Supplementary Table 1).

**Pharmacokinetics, biodistribution and toxicity.** The pharmacokinetics of BAQ ONNs were studied in Sprague–Dawley rats upon intravenous (iv) injection. As shown in Fig. 5a and Supplementary Table 2, the serum concentrations of BAQ ONNs were higher than those of free DiD at the same time points up to 48 h, indicating that the plasma clearance of BAQ ONNs was slower than that of DiD because of the nanoscale characteristics of BAQ ONNs. DiD-labelled BAQ13 NPs were also used to investigate the biodistribution of the NPs in nude mice bearing HT29 tumours. As expected, both in vivo and ex vivo imaging showed that the fluorescence signals of BAQ13 NPs were clearly distinguishable in tumour areas rather than in surrounding normal tissues at 12 and 24 h post injection, indicating the tumour-targeting biodistribution

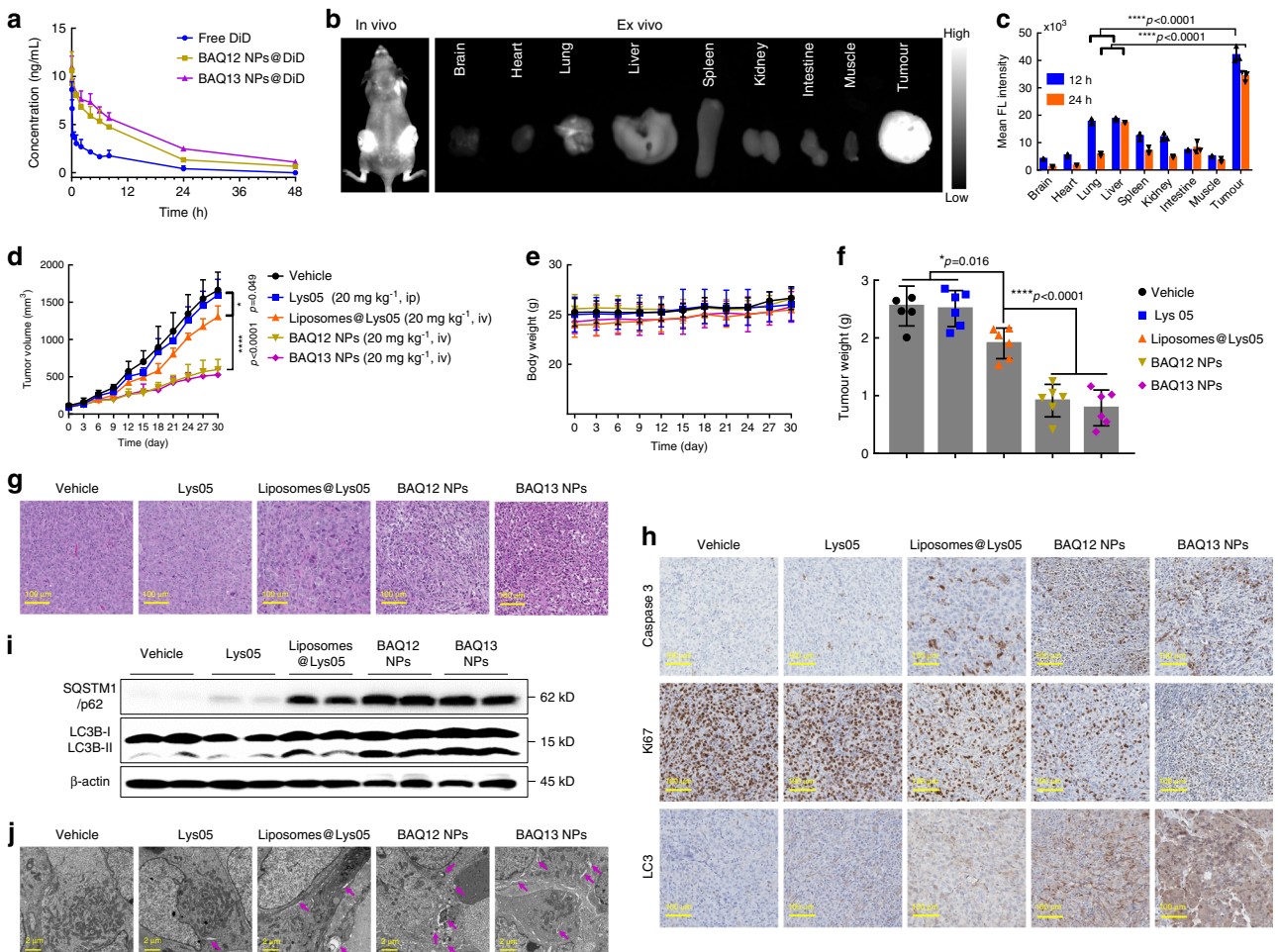

**Fig. 5 The pharmacokinetics, biodistribution and in vivo antitumour effect of BAQ ONNs. a** The plasma concentration-time profiles of DiD-loaded BAQ ONNs and free DiD after intravenous injection; data are mean values ± SD; $n = 3$. **b** In vivo and ex vivo biodistribution of BAQ13 NPs in mice bearing HT29 tumours at 24 h post injection. **c** Quantitative fluorescence intensity of tissues obtained at 12 and 24 h post injection; data are mean values ± SD; $n = 3$. **d** MIA PaCa-2 tumour growth curves in mice that were treated as indicated every 3 days; data are mean values ± SD; $n = 6$ tumours per group. **e** Body weight of mice during the treatment; data are mean values ± SD; $n = 6$ mice per group. **f** Weight of harvested tumours at the end of the treatment; data are mean values ± SD; $n = 6$ tumours per group. **g–j** Representative H&E (**g**), IHC (**h**), immunoblotting (**i**) and TEM (**j**) results of tumours that were harvested at the end of treatments; Blots in **i** were from three individual tumours of each group; purple arrows in **j**: autophagic vesicles; experiments in **g–j** were all repeated three times independently. All statistical $p$ values were calculated by one-way ANOVA with the Tukey's multiple comparison test; *$p < 0.05$; ****$p < 0.0001$.

of BAQ ONNs (Fig. 5b, c and Supplementary Fig. 7a, b). This targeting ability may be due to the relatively high permeabilization of tumour blood vessels, which enables passive accumulation of nanotherapeutics[40]. The free DiD (control) group showed high signals in the lungs, rather than in the tumour sites (Supplementary Fig. 7a, b). In particular, the tumour-to-lung ratio of fluorescence in the BAQ13 NP group was ~4-fold higher than that in the free DiD group. These results demonstrated that BAQ ONNs had a tumour-targeting biodistribution.

We next carried out a haemolysis assay to evaluate the safety of BAQ ONNs. Under physiological conditions, red blood cells were treated with Lys05, BAQ12 NPs or BAQ13 NPs at concentrations of 0.25–1 mg mL$^{-1}$, close to the working concentrations used for the animal treatment study (Supplementary Fig. 7c). Treatment with the control drug Lys05 resulted in a significantly higher haemolytic rate than treatment with BAQ12 NPs or BAQ13 NPs at concentrations above 0.5 mg mL$^{-1}$, indicating the greater safety of BAQ ONNs than Lys05. In the following animal toxicity studies on FVB/N mice, Lys05 treatment was found to cause acute death of mice after iv administration even at a low

concentration of 10 mg kg$^{-1}$; in contrast, BAQ ONN treatment resulted in low mortality and no body weight loss, revealing that BAQ ONNs are safe when administered via iv injection (Supplementary Fig. 7d, e)[24]. Since they did not lead to any death at 40 mg kg$^{-1}$, BAQ13 NPs were better tolerated by the mice than BAQ12 NPs. This result is likely attributable to the high stability and low CAC of BAQ13 NPs. We also used liposomes to encapsulate Lys05 (liposomes@Lys05) and found that this formulation is safe for iv injection (Supplementary Fig. 7e, f). Because autophagy plays an important role in intestinal homoeostasis, intraperitoneal (ip) administration of BAQ12 NPs or BAQ13 NPs, which results in an increased autophagy-inhibiting effect, may cause intestinal disorders and loss of body weight in mice[24,41]. H&E staining of tissue sections and haematologic indexes did not show obvious abnormal alterations in mice treated with 20 mg kg$^{-1}$ BAQ NPs by tail vein, which further suggested that iv administration of BAQ ONNs is well tolerated (Supplementary Fig. 8a–e). Therefore, BAQ ONNs should be administered by iv injection rather than by ip injection for investigation of their advantages in vivo.

**Antitumour effect as single agents in mice**. After proving the safety of BAQ ONNs, we next evaluated the NPs for anticancer efficacy in a pancreatic xenograft model of MIA PaCa-2 cells. NRG mice with MIA PaCa-2 tumours (~100 mm$^3$) were randomly assigned into five groups ($n = 6$): the saline (iv) group, the Lys05 (ip) group, the liposomes@Lys05 (iv) group, the BAQ12 NPs (iv) group and the BAQ13 NPs (iv) group. The mice were then treated every 3 days at a dose of 20 mg kg$^{-1}$. The results in Fig. 5d–f show that the treatment with BAQ12 NPs or BAQ13 NPs significantly decelerated tumour growth without interfering with body weight. The control drug Lys05 did not display a therapeutic effect under this condition, but its nanoformulation, liposomes@Lys05, elicited increased tumour inhibition, which highlights the advantage of nanomedicines in drug delivery. It should also be emphasized that the one-component formulations of self-assembling BAQ12 NPs or BAQ13 NPs were significantly more efficacious than either free Lys05 or nanoformulated Lys05. These findings clearly illustrate the potential advantages of BAQ ONNs with regard to both drug discovery and drug delivery.

To further understand the in vivo effects of BAQ ONNs, tumour tissues were harvested for histological assessment. Dramatic cellular destruction, increased cleaved caspase-3 levels and decreased Ki67 expression were observed in both BAQ ONN groups, suggesting that the tumours treated with BAQ ONNs were inclined to die or to become apoptotic or quiescent (Fig. 5g, h). LC3 expression was increased in both BAQ ONN groups (Fig. 5h); this finding is an essential clue explaining the in vivo autophagy-inhibiting effects of both BAQ ONNs. The subsequent immunoblot analysis further demonstrated that autophagy in tumours was blocked by the BAQ ONNs (Fig. 5i). Additionally, we observed the tissue ultrastructure by TEM and found that BAQ ONN-treated tumours contained more numerous large AVs than the groups of tumours (Fig. 5j). In the assays above, the Lys05 nanoformulation, liposomes@Lys05, also exhibited some effects not observed with the vehicle or free Lys05. However, the effects of liposomes@Lys05 were much weaker than those of either BAQ12 NPs or BAQ13 NPs. These tissue-level results revealed the excellent autophagy-inhibiting effects of BAQ ONNs in vivo.

The therapeutic effects of BAQ ONNs in vivo were further demonstrated in another animal model consisting of mice bearing colon HT29 tumours. Compared with vehicle or Lys05 administration, BAQ ONN administration significantly inhibited tumour growth (Supplementary Fig. 9a, b). And BAQ13 NPs displayed better efficacy than BAQ12 NPs. Interestingly, this result contradicted the results obtained in the in vitro proliferation assay, which demonstrated BAQ12 NPs as being more effective than BAQ13 NPs. This discrepancy could be explained by the differences in self-assembly behaviours and pharmacokinetic profiles between the BAQ ONNs (Figs. and 2g, 5a). Moreover, BAQ13 NPs were also more efficacious than FDA-approved irinotecan at its reported therapeutic dose, while BAQ12 NPs showed effects similar to those of irinotecan[42]. Survival analysis revealed that BAQ13 NP treatment resulted in a significantly longer survival time (median survival of 48 days) than vehicle and irinotecan groups (median survival of 21 or 36 days, respectively) (Supplementary Fig. 9c and Supplementary Table 3). Given their integration of multiple advantages regarding both pharmacodynamic effects and pharmacokinetic profiles, the hybrid BAQ ONNs exhibit enormous potential for cancer treatment in vivo as single agents.

**Dual roles of BAQ ONNs in combination therapy**. Autophagy inhibition-based combination therapy could sensitise tumours to conventional therapeutics, but the current limitation is the insufficient efficacy of autophagy inhibitors[22]. Moreover, the disparate pharmacokinetics and different dosing schedules of drugs used in combination therapy are inconvenient. Given the 30-fold higher anticancer potency of BAQ ONNs than HCQ and their considerable potential to encapsulate additional drugs, BAQ ONNs may be able to address these two pharmacodynamic and pharmacokinetic issues simultaneously. To test this hypothesis, we established a xenograft model with high heterogeneity and a high tumour stroma proportion by using a pancreatic cancer stem cell (PCSC) line from patient-derived pancreatic adenocarcinoma tissue (Fig. 6a, b and Supplementary Fig. 10)[43,44]. The in vitro results proved that BAQ ONNs had similar functions in inhibiting lysosomes and autophagy in PCSCs and therefore exhibited potent proapoptotic and anti-proliferative activities (Fig. 6c–e and Supplementary Fig. 11a–c). The STAT3 inhibitor napabucasin, which can be encapsulated in BAQ13 NPs, was chosen for the combination therapy because it can induce autophagy and synergise with BAQ13 NPs (Fig. 6f and Supplementary Fig. 11d)[45,46]. Mice were randomly divided into 5 groups, including the vehicle (saline) group, the napabucasin group, the BAQ13 NPs group, the mixture (BAQ13 NPs+napabucasin) group and the BAQ13 NPs@napabucasin group (Fig. 6g–i). BAQ13 NPs moderately inhibited tumour growth, while napabucasin itself exhibited no antitumour effect under these conditions. The mixture group did not exhibit an enhanced effect in vivo, although in vitro synergy of napabucasin and BAQ13 NPs was observed. This lack of in vivo effect was probably due to the poor solubility and inefficient delivery of napabucasin[47]. When loaded in BAQ13 NPs (BAQ13 NPs@Napabucasin), the nanoformulated napabucasin achieved a satisfactory antitumour effect by synergising with BAQ13 NPs. Remarkable changes in tumour histology were also observed in the BAQ13 NPs@napabucasin group, in which the cells showed low proliferation activity (Fig. 6j). In addition, none of treatment groups of mice exhibited obvious systemic toxicity (Fig. 6i and Supplementary Fig. 11e). To further verify the ability of BAQ13 NPs to deliver napabucasin, we performed another imaging study on the PCSC model by using DiD-labelled BAQ13 NPs@napabucasin. The results showed obvious accumulation of NPs in tumour sites rather than normal organs (Fig. 6k, l and Supplementary Fig. 11f). These interesting results demonstrate that BAQ13 NPs can function not only as therapeutic agents but also as delivery carriers in combination therapy; therefore, they show promise for improving cancer treatment.

## Discussion

Based on an ONN strategy and the principles of pharmacophore hybridisation and molecular self-assembly, we developed the self-delivering new-chemical entities, BAQ ONNs. These entities were equipped with enhanced abilities to induce lysosomal disruption, lysosomal dysfunction and autophagy blockade in addition to improved properties for drug delivery and tumour-targeted biodistribution; thus, they exhibited significant anticancer efficacy both in vitro and in vivo. Strikingly, we found that the simple BAQ13 NPs showed high drug-loading efficiency and could potently synergise with and deliver an additional drug, thus showing promise for application in combination therapy.

In contrast to conventional NPs, which typically have an active pharmaceutical ingredient (API) content of less than 20% and are complicated to synthesise, BAQ ONNs have a 100% API content and are easy to synthesise and scale up. Since they are non-prodrug chemical entities, they are also superior to emerging one-component prodrug NPs. All these advantages will greatly facilitate their translation into clinical trials. To the best of our knowledge, this is an important attempt to extend nanotechnology into the design of new-chemical entities. A seamless connection between drug discovery and nanotechnology-assisted drug delivery

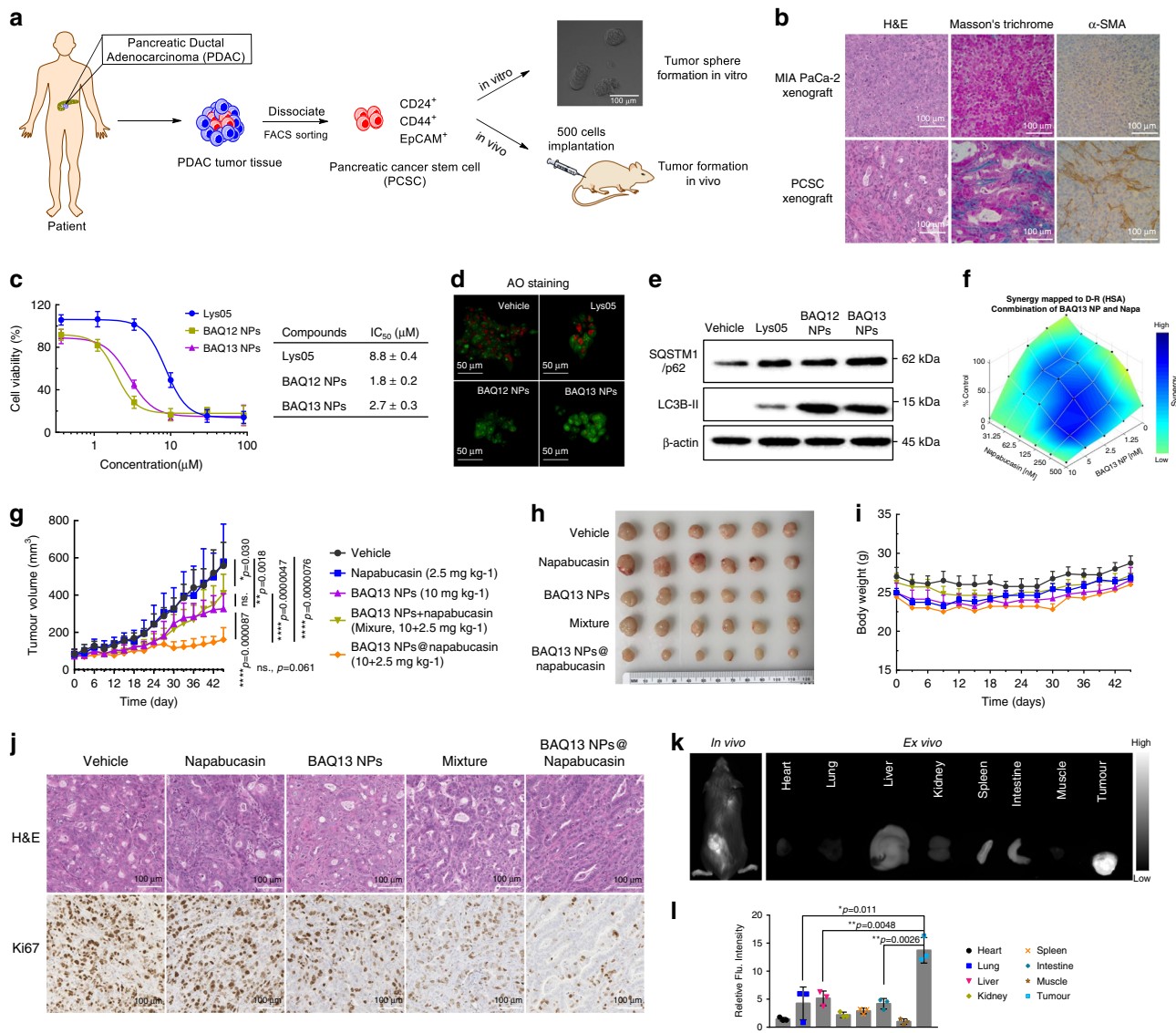

**Fig. 6 BAQ ONNs have dual roles in the combination treatment. a** Establishment of the patient-derived pancreatic cancer stem cell (PCSC) model. **b** Histological analysis showing the high-level stroma of PCSC tumours; experiments were repeated three times independently. **c** Viability curves of PCSCs that were treated for 48 h and the IC$_{50}$ values; $n = 3$ independent experiments. **d** AO staining to show the LMP of PCSC that were treated for 12 h; experiments were repeated three times independently. **e** Immunoblotting analysis of autophagy proteins in PCSC that were treated as indicated (2.5 μM, 48 h); experiments were repeated three times independently. **f** Synergistic effect of BAQ13 NPs and napabucasin (48 h). **g** Tumour growth curves in subcutaneous PCSC xenograft model within the indicated treatments every 3 days; data are mean values ± SD; $n = 10$ tumours per group. **h** Images of tumours that harvested at end of treatment. **i** Mice body weight changes during the treatment; data are mean values ± SD; $n = 5$ mice per group. **j** Representative images of PCSC tumour sections; experiments were repeated three times independently. **k** In vivo and ex vivo fluorescence imaging of BAQ13 NPs co-loading with napabucasin and DiD in the PCSC model at 48 h post intravenous injection (10 mg kg$^{-1}$). **l** Quantitative fluorescence intensity of tissues in **k**, data are mean values ± SD; $n = 3$ mice per group. All statistical $p$ values were calculated by the two-tailed Student's $t$ test. ns not significant; *$p < 0.05$; **$p < 0.01$; ****$p < 0.0001$.

will enable researchers to develop increasingly advanced nanomedicines with a wide range of therapeutic and commercial benefits for cancer targeting.

## Methods

**Preparation and characterisation of BAQ ONNs**. NPs were prepared through the re-precipitation method. Briefly, BAQ derivatives in methanol were added drop-wise into MilliQ water while stirring for 5 min (volume ratio, 1:10), and then homogenous NPs were obtained after rotary evaporation (40 °C, 20 min), followed by the characterisation with Zetasizer Nano ZS (Malvern). The TEM samples were prepared by dropping 0.5 mM NPs on carbon square mesh and dried naturally, which were then observed under the Talos L120C TEM (FEI) at an accelerating voltage of 80 kV. To determine the drug content in nanoformulations, the prepared drug-loaded NPs were cut off by centrifugal filter (ultracel-10 kDa, Millipore), and

the absorbance of filtrate (diluted with DMSO, 1:10, volume ratio) was measured for calculation of drug concentrations.

**Cell viability, cell growth and colony formation**. Cell viability was assessed by the MTS assay. Briefly, cells in 96-well plates (4000 cell per well) were treated as indicated, followed by the incubation with MTS regents for 4 h. OD values (490 nm) were determined via a microplate reader. Results were shown as the average cell viability calculated from the formula of $[(\text{OD}_{treat} - \text{OD}_{blank})/(\text{OD}_{control} - \text{OD}_{blank}) \times 100\%]$. Drug combination data were analyzed by Combenefit 2.02. In cell growth assay, cells in 6-well plates (50,000 cell per well) were treated as indicated and were counted manually every 24 h. Colony formation assay was also performed on 6-well plates with a starting density of 1000–2000 cells per well. After incubated as indicated for 10–20 days, cells were washed with PBS and stained with the solution of crystal violet and methanol for 20 min.

**Apoptosis and caspase-3/7 activity**. Cell apoptosis was measured using FITC-Annexin V/PI Apoptosis kit (AnaSpec). Briefly, the treated cells were stained according to the manufacturer's instructions and were detected on a BD FACSCanto II flow cytometer. Data were analyzed by FlowJo 7.6.1. In caspase 3/7 activity assay, cells in 96-well plates (10,000 cells per well) were treated as indicated, followed by adding AMC caspase-3/7 assay kit (50 µL per well, AnaSpec). The fluorescence intensity ($\lambda_{ex} = 356$ nm, $\lambda_{em} = 442$ nm) was recorded by a microplate reader.

**Cell uptake and deacidification**. For cell uptake, lysosomes were labelled with Alexa Fluor 488-dextran (10 kDa, 100 µg mL$^{-1}$, Thermo Fisher) for 36 h, followed by incubation with DiD-loaded BAQ NPs (10 uM, 1:10, mass ratio) for 2 h. In lysosomal deacidification analysis, cells were treated for 2 h and incubated with LysoTracker Red (100 nM, Thermo Fisher) for 1 h. Cell images were obtained using a Zeiss Confocal Microscope and analyzed by Zen 2.3 and ImageJ 1.51 s.

**Lysosome integrity**. Lysosomal integrity was measured in living cells by using the AO (Thermo Fisher) or Alexa Fluor 488-dextran (10 kDa) staining. For AO staining, the treated cells were incubated with AO (2 µg mL$^{-1}$) for 1 h. For dextran staining, the dextran-loaded cells were exposed to treatments for 12 h. Images were captured under a Zeiss Confocal Microscope and analyzed by Zen 2.3 and ImageJ 1.51 s.

**LC3B-GFP imaging**. Cells in a 96-well plate (5000 cell per well) were transfected by the autophagy sensor LC3B-GFP (Thermo Fisher) for 12 h. After treated as indicated for 4 h, cells were visualised by a fluorescence microscope (Olympus). The puncta per well were quantified using ImageJ 1.51 s.

**Lysosome isolation and cathepsin release**. Lysosomes were isolated using a Lysosome Enrichment Kit (Thermo Fisher) according to the manufacturer's protocol. The equal portions of isolated lysosomes were incubated as indicated for 12 h at 37 °C and then was centrifuged at $15,000 \times g$ for 30 min at 4 °C to pellet intact lysosomes. The release of cathepsin B into the supernatant was determined (Ex = 380 mm, Em = 460 mm) after a 2 h incubation with 200 µM fluorogenic Cathepsin B Substrate III (Z-Arg-Arg-AMC).

**Haemolysis**. Red blood cells (2%) in PBS (10 mM, pH 7.4) were incubated with NPs for 4 h at 37 °C. After centrifugation at $500 \times g$ for 5 min, the extent of haemolysis was spectrophotometrically determined according to the amount of haemoglobin in supernatants (540 nm). The haemolysis assay was used to assess the pH-dependent detergence ability and toxicity of NPs.

**Western blot**. The cell or tumour samples were lysed with RIPA Buffer (Thermo Fisher). After centrifugation at 4 °C (15 min, $12,000 \times g$), the concentrations of proteins in supernatant and determined by Bradford Protein Assay dye (Bio-Rad). Immunoblotting was performed routinely and were developed using a Chemi-Doc$^{TM}$ MP imaging system.

**TEM of cells and tumour tissue**. MIA PaCa-2 cells in 8-well slide plates (30,000 cell per well, Lab-Tek) were treated as indicated for 48 h. The freshly harvested tumours were cut into 1 mm$^3$ pieces. Samples were fixed with the 0.1 M cacodylate buffer containing 2.5% glutaraldehyde plus 2% paraformaldehyde, and transferred onto carbon square mesh, followed by observation under Talos L120C TEM.

**RNA-seq**. Total RNA was extracted by the RNeasy Mini Kit (Qiagen, Germany) from the treated MIA PaCa-2 cells (5 µM, 24 h). Samples were submitted to the UC Davis Comprehensive Cancer Centre's Genomics Shared Resource (GSR) for RNA-Seq analysis. Stranded RNA-Seq libraries were prepared from 100 ng total RNA using the NEBNext Ultra Directional RNA Library Prep Kit (New England Bio-Labs). Subsequently, libraries were combined for multiplex sequencing on an Illumina HiSeq 4000 System ($2 \times 150$ bp, paired-end, $>20 \times 10^6$ reads per sample). The data of normalised genes read counts were analyzed using fold change and t test. The Differentially expressed genes (DEGs) were collected for the signalling pathways enrichment by Funrich software 3.1.3. The gene sets were from MSigDB database (Broad Institute). GSEA was performed using GSEA version 3.0 in KEGG gene sets category online, with the following parameters: $n = 1000$ permutations, where $p$ adjust $< 0.05$, and FDR $< 0.05$ were considered significant.

**qPCR**. The total RNA was isolated using the TRIZOL reagent (Invitrogen) and the phenol-chloroform extraction method. The cDNA was synthesised using Super-Script II reverse transcriptase (Invitrogen) with 2 µg of total RNA in a 20 µL reaction. The resulting cDNA was diluted 1:20 in nuclease-free water and 4 µL was used per qPCR reaction with triplicates. qPCR was carried out using Power SYBR Green PCR Master Mix (Thermo Fisher) on a CFX96 Real-Time PCR Detection System (Bio-Rad) including a non-template negative control. Amplification of GAPDH was used to normalise the level of mRNA expression. The primer sequences were listed in Supplementary Table 4.

**Lipidomics**. MIA PaCa-2 cells were treated with compounds (2.5 µM) for 48 h, and 1.5 million cells in each group were collected to prepare the samples routinely for RPLC-QTOF analysis. The samples were run on a Vanquish UHPLC System, followed by data acquisition using a Q-Exactive HF Hybrid Quadrupole-Orbitrap Mass Spectrometer. The LC–MS data were processed using MS-DIAL 3.70. Statistical analysis was done by first normalising data using the sum of the knowns, or mTIC normalisation, to scale each sample. Normalised peak heights were then submitted to R 3.5.1 for statistical analysis. ANOVA analysis was performed with FDR correction and post hoc testing.

**Animal model**. To establish the subcutaneous xenograft models, $5 \times 10^6$ of MIA PaCa-2 cells, $2 \times 10^6$ of HT29 cells or $2 \times 10^4$ PCSCs suspended with Matrigel (Corning) and PBS mixture (1:1, volume ratio) were injected subcutaneously into the right flank of nude mice or NRG mice, respectively.

**In vivo treatment schedule**. The NRG mice bearing MIA PaCa-2 xenograft tumours (~100 mm$^3$) were randomised into 5 groups ($n = 6$), and then were subjected to iv injection every 3 days as indicated. For HT29 xenograft model, six groups of nude mice ($n = 6$) with 100 mm$^3$ of tumours were administrated every 3 days with vehicle (saline, iv), Lys05 (ip), BAQ12 NPs (iv), BAQ13 NPs (iv), Irinotecan (ip), respectively. The treatment on HT29 model was stopped on Day 24, and then the mice survival in each group was recorded, in which the mouse with a tumour larger than 1000 mm$^3$ was considered dead. For the co-delivery study, NRG mice ($n = 5$) bearing PCSC tumours were treated with vehicle (saline, iv), napabucasin (ip), BAQ13 NPs (iv), BAQ13 NPs+napabucasin (iv and ip, respectively) and BAQ13 NPs@napabucasin (iv) every 3 days. The tumour volume and body weight were recorded before drug administration every time. At the end of the treatment, mice were sacrificed and the tumours were collected for further analysis.

**Statistics**. Statistical analysis was performed using GraphPad Prism 7.0. Data are presented as mean values ± SD, $n =$ biological replicates or independent nanoparticle sample replicates. One-way ANOVA with the Tukey's multiple comparison test or two-tailed Student's $t$ test was used to calculate the $p$ value as noted in each figure legend. ns not significant; $*p < 0.05$; $**p < 0.01$; $***p < 0.001$; $****p < 0.0001$.

**Reporting summary**. Further information on research design is available in the Nature Research Reporting Summary linked to this article.

## Data availability

The RNA-seq data have been deposited in the Gene Expression Omnibus (GEO) database under the accession code GSE154323. The Source data underlying Figs. 2a–g, 3e–g, i, k, 4d–g, I, j, 5a, c–f, i, 6c, e, g, i, l and Supplementary Figs. 2a, c, 3a, b, d–f, 4b, e, 5c–f, 6, 7b–f, 8b–e, 9a–c, 11c, d are provided as a Source data file. All the other data supporting the findings of this study are available within the article and its Supplementary Information files and from the corresponding author upon reasonable request. A reporting summary for this article is available as a Supplementary Information file. Source data are provided with this paper.

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

## Acknowledgements

We thank the support from NIH/NCI (R01CA199668, R01CA232845), NIH/NIDCR (1R01DE029237), NIH/NICHD (R01HD086195), UC Davis Comprehensive Cancer Centre Support Grant (CCSG) awarded by the National Cancer Institute (NCI P30CA093373), the Training Grant in Oncogenic Signals and Chromosome Biology T32 CA108459 (awarded to L.S.), and the International Postdoctoral Exchange Fellowship Programme in China (awarded to Z.M.). The authors also appreciate the access to the Molecular Pharmacology Shared Resources and Genomics Shared Resource funded by the CCSG.

## Author contributions

Z.M., Y.L. and M.L. conceived the initial idea and designed the compounds. Z.M. synthesised compounds, prepared NPs and carried out the characterisation of chemicals and NPs. J.L., K.L., Z.M., M.R. and B.J. conducted in vitro anticancer evaluation and cell imaging studies. K.L., J.L. and D.Z. contributed to the western blot experiments. D.Z., and J.L. performed the RNA-seq and qPCR experiments. M.R., K.L., M.S. and O.F. carried out the lipidomics studies. B.J., M.R. and Z.M. performed the pharmacokinetics study. Z.M., J.L., K.L. and B.J. conducted the in vivo evaluation. Z.M. drafted and revised the paper. A.L., S.U., L.-N.S., C.-D.S. and M.R. edited the paper. T.L. assisted with the biological experiments. Y.D., S.U. and M.R. established the cancer stem cell model. Y.L. supervised the entire project and the overall paper preparation and revision.

## Competing interests

Y.L. and Z.M. are the co-inventors on a pending patent application on the hybrid compounds and the resulting nanoformulations. The remaining authors declare no competing interests.

## Additional information

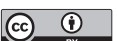

