## [Peer Review File · Nature Communications]

Reviewers' comments:

Reviewer #1 (Expertise: Autophagy, cancer, Remarks to the Author):

This is an extremely rigorous and quite extensive analysis of the capacity of a newly developed nanomolecular based chemical entity to act as an antitumor drug, ostensibly through interference with autophagy via lysosomal disruption. The authors provide pharmacokinetic studies as well as evidence of antitumor activity both in cell culture and tumor bearing animal studies.

There is one major concerns, which is the reliance on MCF10A cells as an indication of drug selectivity. These cells do not actually reflect "normal" tissue. The authors should include studies with e.g. normal human fibroblasts and possibly bone marrow cells in order to more firmly establish that their molecular entity is highly selective for tumor cells.

Other points

1. The manuscript does not explain why an increase in LC3B-II levels and in SQSTM1/p62 (note the current terminology) are indicative of interference with autophagy. This would readily be apparent to those with expertise in the field but not to the general scientific readership. Also, more care should be taken with regard to the use of the terminology, "autophagic flux". Simply analyzing LC3I to II conversion does not actually measure this aspect of autophagy. This would require the inclusion of an autophagy inhibitor, which would not be feasible experimentally in the current work.
2. It is suggested that the data relating to FOXO3a be removed since the presumption that FOXO3a might be responsible for sensitization to apoptosis is not actually examined in this work.
3. Lines 292-299. Again, the conclusion relating to "autophagic flux" need to be toned down; the data in these Figures do suggest alterations in autophagy but not necessarily autophagic flux. Furthermore, the Western blotting data in Figure 5i do not provide convincing support for any significant alterations in autophagy.
4. Lines 329-333. Again, please consider appropriate modifications to the discussion of "autophagic flux" based on the data in Figures 6 and Supplementary Figures 9.

Reviewer #2 (Expertise: Drug design, med chem/drug discovery, Remarks to the Author):

Overall, this is a strong manuscript describing the design, synthesis, and assessment of hybrid drugs that simultaneously address potency and delivery properties. Specifically, the structural features of lysomotrophic chloroquine were combined with a long aliphatic chain, which had been used in the design of lysomotrophic detergents, to endow detergent properties and to allow self-assembly for nanoparticle formation of lysomotrophic chloroquine. Such a design led to improved potency and delivery properties; it also allowed for encapsulation of other payloads for co-delivery of additional active ingredient(s). The authors have conducted a large number of assays including biochemical and cellular studies as well as animal models of several cancer types. The results do support the idea that the newly design hybrid drugs are more potent and have improved properties. However, the manuscript also needs major improvements before publication is recommended. First, the manuscript needs some serious and extensive editing to improve the English. Second, the title is misleading. For all practical purpose, the manuscript describes the design of hybrid drugs, which have improved properties. The work uses some techniques in nanotechnology; however, it is not "nanotechnology directed"

discovery. The title might be catchy, but does not reflect the essence or the bulk of the work. Third, the work is described as a novel “platform” approach. However, this reviewer fails to see this aspect. Lysosomal targeting is very unique and allows the incorporation of detergent properties for the “drug” component. However, this is somewhat idiosyncratic for lysosomes and not generally applicable to other targets. Fourth, abstracts are meant to be informative. It is very hard to understand what the work is about after reading the abstract. This might be because of the focus on “platform” technology, which it is not.

The work and results described in the manuscript are strong and can stand on their own merit without superlatives and catchy phrases, which actually become distracting.

Reviewer #3 (Expertise: Stimuli response nanoparticles, drug delivery, Remarks to the Author):

The present publication deals with new chemical entities aimed to block cancer autophagy, a relevant and critical resistance mechanism during multimodal cancer treatment. Furthermore, the authors provide experimental data which may stimulate development of improved anticancer agents or new combinations of drugs with different mode of action. They suggest a “One-component New-chemical-entity Nanomedicine (ONN) concept” and utilize nanotechnological methodologies for improving efficacy and potency of established pharmacophores. Based on a validated and optimized chloroquine derivative, they designed new, micelle forming chloroquine derivatives by chemical conjugation of a lipophilic tail molecules (MSDH) to the known lysosome inhibitor Lys05. Furthermore, by testing the growth inhibitory action in a series of human cancer cell lines, an improvement of the IC50 for growth inhibition was demonstrated after 24 and 48 hours of treatment (Fig. 2a). At physiological pH (7.4) no safety improvement of the selected BAQ12 and BAQ13 NP formulations over Lys05 positive control is provided (Fig. 2c). The authors did not consider CMC (critical micelle concentration) as a parameter needed for characterization of the nanoformulation. Experimental data on lysosomal targeting and disruption are convincing. Both the nanoformulation of Lys05 and the blank Lys05 interfere with autophagy mechanisms. The different effect levels (Fig. 3) may reflect the slightly different IC50. Additional validation of the growth inhibition is provided by the demonstration of treatment-associated gene expression and apoptosis induction. Here, differences between the nanoformulation of Lys05 and the positive control Lys05 are provided. However, the difference is certainly due to the inappropriate concentration of the Lys05 positive control. If a working concentration close to the IC50 would be used, an expression of apoptosis marker can be expected (Fig. 4 i, j). With respect to the in-vivo examination, NIR bioimaging studies, PK blood sampling and tumor growth studies are provided (Fig. 5-6). Again, both the nanoformulation and the blank of Lys05 exert inhibitory activity. The significant higher growth inhibition with the nanoformulation of Lys05 is associated with a decrease of the body weight, an indicator of toxicity. The positive control Lys05 did not impair body weight gain compared to PBS controls which may imply better safety.

General remarks:

The topic of the manuscript is of general interest in the cancer research field. All experiments were conducted properly and conclusively presented. However, the data do not support the idea of the authors of a new paradigm or platform for nanodrugs. The improvements in drug efficacy are modest and accompanied with increased toxicity. There is no convincing data on a fundamentally different profile of the nanoformulated pharmacophore compared with the Lys05 compound. Basic information such as CMC or stability of the formulation is missing. So far, no added value of the nano aspect can be seen.

Response to Reviewers' comments

Reviewer #1 (Expertise: Autophagy, cancer, Remarks to the Author):

This is an extremely rigorous and quite extensive analysis of the capacity of a newly developed nanomolecular based chemical entity to act as an antitumor drug, ostensibly through interference with autophagy via lysosomal disruption. The authors provide pharmacokinetic studies as well as evidence of antitumor activity both in cell culture and tumor bearing animal studies.

There is one major concern, which is the reliance on MCF10A cells as an indication of drug selectivity. These cells do not actually reflect "normal" tissue. The authors should include studies with e.g. normal human fibroblasts and possibly bone marrow cells in order to more firmly establish that their molecular entity is highly selective for tumor cells.

Re: We thank the reviewer for the constructive comments. As suggested, we tested the cell viability on three non-cancerous cell lines, including IMR-90 cells (human lung fibroblast), NIH/3T3 cells (mouse embryo fibroblast) and bone marrow cells. As shown in Supplementary Fig. 6, BAQ12 and BAQ13 showed more than 6 μM of IC_{50} values on these non-cancerous cell lines, which are about 3 times higher than that of tumour cells. Therefore, the new BAQ molecular entities exhibited relative high selectivity to tumour cells.

Other points:

1. (1) The manuscript does not explain why an increase in LC3B-II levels and in SQSTM1/p62 (note the current terminology) are indicative of interference with autophagy. This would readily be apparent to those with expertise in the field but not to the general scientific readership. (2) Also, more care should be taken with regard to the use of the terminology, "autophagic flux". Simply analyzing LC3I to II conversion does not actually measure this aspect of autophagy. This would require the inclusion of an autophagy inhibitor, which would not be feasible experimentally in the current work.

Re: Thanks very much for the helpful suggestions. **(1)** The relevant explanation about the working principle of autophagy markers was added the revised manuscript (**Lines 162-169**).¹ "During autophagy, the cytosolic form of LC3 (LC3-I) is converted into the lipid modified form (LC3-II), which is then recruited to the autophagosomal membrane. Meanwhile, the autophagy substrate SQSTM1/p62 protein is degraded via selective incorporation into autophagosomes. Therefore, increased levels of both LC3-II and SQSTM1/p62 should be observed when autophagy is inhibited, while increased LC3-II levels and decreased SQSTM1/p62 levels should be observed if autophagy is activated." We also corrected the p62 terminology by using SQSTM1/p62 throughout the revised manuscript. **(2)** As suggested by the reviewer, we revised the contents relating to the terminology of "autophagic flux", which was replaced with the exact

description of “autophagy” or “autophagy process”. In order to confirm the autophagy-inhibiting effect of BAQ derivatives, another autophagy inhibitor BfA1 was included as a positive control in the western blotting assay (Fig. 3f).² As similar as BfA1, BAQ ONNs increased the levels of LC3-II and SQSTM1/p62 in MIA PaCa-2 cells, which clearly indicated BAQ ONNs could inhibit autophagy in cells.

Revised Fig. 3f Immunoblotting analysis of cells that were treated as indicated for 24 h.

2. It is suggested that the data relating to FOXO3a be removed since the presumption that FOXO3a might be responsible for sensitization to apoptosis is not actually examined in this work.

Re: Thanks for this suggestion. We have removed the paragraphs and data related to FOXO3a.

3. Lines 292-299. Again, the conclusion relating to "autophagic flux" need to be toned down; the data in these Figures do suggest alterations in autophagy but not necessarily autophagic flux. Furthermore, the Western blotting data in Figure 5i do not provide convincing support for any significant alterations in autophagy.

Re: We thank the reviewer very much for the great suggestions. We toned down the relevant conclusions and used the term of “autophagy” to replace “autophagic flux”, which were updated to **Lines 280-288** due to the revision. To further determine autophagy changes *in vivo*, we re-ran the animal study and revised **Fig. 5i** (shown below, **Page 7**) by optimizing western blotting conditions. Both LC3B-II and SQSTM1/p62 were obviously increased by treatments with BAQ12 NPs or BAQ13 NPs, which indicates that the autophagy process in tumours was inhibited by BAQ ONNs. In this new animal study, the Lys05 nanoformulation, liposomes@Lys05, was enclosed as another control to better characterize the advantages of BAQ ONNs with regard to both drug discovery and drug delivery. BAQ ONNs were more potent to impair the autophagy *in vivo* than either the free Lys05 or nanoformulated Lys05.

4. Lines 329-333. Again, please consider appropriate modifications to the discussion of "autophagic flux" based on the data in Figures 6 and Supplementary Figures 9.

Re: Thanks for the comments very much. We modified the discussion related to "autophagic flux" as suggested. This part was updated to **Lines 311-315** that refers to **Fig. 6** and **Supplementary Fig. 11**. We also revised throughout the manuscript to ensure that all the relevant contents about "autophagic flux" have been corrected.

Reviewer #2 (Expertise: Drug design, med chem/drug discovery, Remarks to the Author):

1. Overall, this is a strong manuscript describing the design, synthesis, and assessment of hybrid drugs that simultaneously address potency and delivery properties. Specifically, the structural features of lysomotrophic chloroquine were combined with a long aliphatic chain, which had been used in the design of lysomotrophic detergents, to endow detergent properties and to allow self-assembly for nanoparticle formation of lysomotrophic chloroquine. Such a design led to improved potency and delivery properties; it also allowed for encapsulation of other payloads for co-delivery of additional active ingredient(s). The authors have conducted a large number of assays including biochemical and cellular studies as well as animal models of several cancer types. The results do support the idea that the newly design hybrid drugs are more potent and have improved properties.

Re: We thank the reviewer very much for the positive comments.

2. However, the manuscript also needs major improvements before publication is recommended. (1) First, the manuscript needs some serious and extensive editing to improve the English. (2) Second, the title is

misleading. For all practical purpose, the manuscript describes the design of hybrid drugs, which have improved properties. The work uses some techniques in nanotechnology; however, it is not “nanotechnology directed” discovery. The title might be catchy, but does not reflect the essence or the bulk of the work. (3) Third, the work is described as a novel “platform” approach. However, this reviewer fails to see this aspect. Lysosomal targeting is very unique and allows the incorporation of detergent properties for the “drug” component. However, this is somewhat idiosyncratic for lysosomes and not generally applicable to other targets. (4) Fourth, abstracts are meant to be informative. It is very hard to understand what the work is about after reading the abstract. This might be because of the focus on “platform” technology, which it is not.

Re: Thanks very much for these valuable suggestions. (1) According to your suggestions, we had spent a lot of time in working with the native English-speaking co-authors to edit the manuscript and improve the English. Furthermore, we used the service from Springer Nature on English Language Editing before resubmission. The editing certificate is as follows:

(2) As suggested, we changed the title to “Pharmacophore hybridization and nanoscale assembly to discover new self-deliverable lysosomotropic chemical entities for cancer therapy”. This title could better summarize our work. (3) Based on the comments, we deleted the relevant words about the “platform” approach. Lysosomal targeting is exactly a good model to verify our ONN strategy. As we have observed the feasibility of this strategy, we are expanding this strategy to other drug targets, such other organelles and tumour-associated proteins. This work is ongoing. (4) We have rewritten the abstract part, and the new version is focused on summarizing our work here. We also deleted the terms regarding the “platform” technology.

3. The work and results described in the manuscript are strong and can stand on their own merit without superlatives and catchy phrases, which actually become distracting.

Re: Thank you for the comments. We have toned down our claims and revised throughout the manuscript.

Reviewer #3 (Expertise: Stimuli response nanoparticles, drug delivery, Remarks to the Author):

1. The present publication deals with new chemical entities aimed to block cancer autophagy, a relevant and critical resistance mechanism during multimodal cancer treatment. Furthermore, the authors provide experimental data which may stimulate development of improved anticancer agents or new combinations of drugs with different mode of action. They suggest a “One-component New-chemical-entity Nanomedicine (ONN) concept” and utilize nanotechnological methodologies for improving efficacy and potency of established pharmacophores. Based on a validated and optimized chloroquine derivative, they designed new, micelle forming chloroquine derivatives by chemical conjugation of a lipophilic tail molecules (MSDH) to the known lysosome inhibitor Lys05. Furthermore, by testing the growth inhibitory action in a series of human cancer cell lines, an improvement of the IC₅₀ for growth inhibition was demonstrated after 24 and 48 hours of treatment (Fig. 2a). At physiological pH (7.4) no safety improvement of the selected BAQ12 and BAQ13 NP formulations over Lys05 positive control is provided (Fig. 2c).

Re: Thank you very much for the important comments. By using the low concentration of compounds (50 μ M), the haemolysis test in **Fig. 2c** is used to show the pH-responsive biomembrane disruption ability of BAQ ONNs, rather than a safety evaluation (**Lines 93-94**).³ In this case, none of Lys05 and BAQ ONNs had haemolytic effects at physiological pH (7.4). BAQ ONNs caused obvious haemolysis under simulated lysosomal conditions (pH 4.0~5.5). This is because they acquired the detergent and the specific pH-responsive membrane disruption ability in acidic environments. Without this ability, Lys05 did not result in much haemolysis in the whole pH range at this low concentration. To further emphasize the safety of BAQ ONNs, we evaluated their hemolytic ability at high concentrations (0.25-1.0 mg/mL). These concentrations could better reflect the reality of the practical formulation used in animal study. As a result, BAQ12 NPs and BAQ13 NPs showed the significantly lower hemolysis than Lys05 at physiological pH (7.4), suggesting a safety improvement (**Supplementary Fig. 7c, Lines 247-252**). In fact, iv administration of Lys05 was not appropriate to mice because it caused acute death of mice even at a low concentration of 10 mg/kg; in contrast, BAQ ONN treatment resulted in low mortality and no body weight loss, revealing that BAQ ONNs are safe when administered via iv injection. (**Supplementary Fig. 7d, e, Lines 252-264**). H&E staining of tissue sections and haematologic indexes did not show obvious abnormal alterations in mice treated with 20 mg/kg BAQ NPs by tail vein, which further suggested that iv administration of BAQ ONNs is well tolerated (**Supplementary Fig. 8**).

Revised Supplementary Fig. 7 (c) Concentration-dependent hemolysis induced by the corresponding treatments at physiological pH, **(d)** Survival of FVB/n mice that were iv injected with the corresponding agents every two days. **(e)** Body weight of mice that were treated every two days as indicated.

2. The authors did not consider CMC (critical micelle concentration) as a parameter needed for characterization of the nanoformulation.

Re: We had the CMC data in **Fig. 2h** that were described as the critical aggregation concentration (CAC) in **Lines 129-130**. The CACs are 0.44 μ g/mL (0.76 μ M) and 0.15 μ g/mL (0.25 μ M) for BAQ12 NPs and BAQ13 NPs, respectively.

3. Experimental data on lysosomal targeting and disruption are convincing. Both the nanoformulation of Lys05 and the blank Lys05 interfere with autophagy mechanisms. The different effect levels (Fig. 3) may reflect the slightly different IC₅₀. Additional validation of the growth inhibition is provided by the demonstration of treatment-associated gene expression and apoptosis induction. Here, differences between the nanoformulation of Lys05 and the positive control Lys05 are provided. However, the difference is

certainly due to the inappropriate concentration of the Lys05 positive control. If a working concentration close to the IC₅₀ would be used, an expression of apoptosis marker can be expected (Fig. 4 i, j).

Re: We thank the reviewer very much for these insightful comments. In addition to having the similar autophagy-inhibiting effect with Lys05, BAQ12 NPs and BAQ13 NPs also exhibited enormous potential to directly induce lysosomal membrane permeabilization (LMP) as they can act as lysosomotropic detergents (**Fig. 2c, 3c, and 3d**). Furthermore, BAQ12 NPs and BAQ13 NPs also acquired a strong proton-sponging effect that can induce lysosomal swelling and dysfunction (**Fig. 2d, 3j, 3k and 4a-e**). During LMP or lysosomal swelling, the proteolytic enzymes (i.e. cathepsins) in lysosomes are released into cytoplasm, which is an important trigger of apoptosis.⁴ Therefore, BAQ12 and BAQ13 NPs capable of these multiple functions, were more potent in inducing apoptosis over Lys05 whose main function is autophagy inhibition. According to your suggestions, we increased the concentration of Lys05 to 10 μ M in caspase 3/7 activity assay and to 15 μ M in apoptosis assay. As shown in **Fig. 4 i and 4j**, Lys05 increased the apoptotic signals in concentration-dependent manner, but its effect at the high concentration close to the IC₅₀ was still milder than those of the low concentrations of BAQ ONNs (**Lines 223-230**). Therefore, it could be demonstrated that cancer cells are more sensitive to the multifunctional BAQ ONNs compared to the autophagy inhibitor Lys05.

Revised Supplementary Fig. 4 (i) Caspase 3/7 activity in MIA PaCa-2 and HT29 cells that were treated as indicated for 6h and 12h, respectively.

(j) Percentage of apoptotic population of MIA PaCa-2 (upper) and HT29 (bottom) cells that were treated as indicated for 24h.

(4) With respect to the in-vivo examination, NIR bioimaging studies, PK blood sampling and tumor growth studies are provided (Fig. 5-6). Again, both the nanoformulation and the blank of Lys05 exert inhibitory activity. The significant higher growth inhibition with the nanoformulation of Lys05 is associated with a decrease of the body weight, an indicator of toxicity. The positive control Lys05 did not impair body weight gain compared to PBS controls which may imply better safety.

Re: Thank you very much for the valuable comments. Because iv injection of Lys05 can cause acute death of mice, we utilized ip injection as the same administration method for all the treatment groups to compare their therapeutic effects in the animal study involved in previous **Fig. 5**. We did see the body weight decrease in the groups treated with BAQ ONNs under this condition. As autophagy plays an important role in intestinal homeostasis, the locally ip injection of BAQ ONNs with the high autophagy-inhibiting effect, was possible to cause intestinal disorders and loss of body weight.^{5,6} So ip injection was not a good way to fully demonstrate the advantages of the new BAQ entities *in vivo*. Considering iv administration of BAQ ONNs was better tolerated by the mice (**Supplementary Fig. 7d,e and Supplementary Fig. 8**), we followed the reviewer's suggestions and redesigned the animal study (**Fig. 5d-j, Lines 265-276**), in which mice were treated with BAQ ONNs through tail vein. To better show the difference of the free drug Lys05 and the nanoformulated BAQ ONNs, the dosing frequency was decreased from every two days to every

three days. In addition, we also added another control, a Lys05 nanoformulation (liposomes@Lys05) that can be iv injected safely. The new results in the revised Fig. 5 showed the free drug Lys05 had very low treatment efficiency and showed improved antitumour effect when loaded in liposomes. Compared to either free or nanoformulated Lys05, the self-assembling BAQ ONNs displayed significantly better antitumour effects *in vivo* without interfering the body weight of mice, thus indicating the comprehensive advantages of BAQ ONNs with regard to both drug discovery and drug delivery.

Revised Fig. 5 (d) The MIA PaCa-2 tumour growth curves in mice that were treated as indicated every three days, n=6. **(e)** Body weight of mice during the treatment. **(f)** Weight of harvested tumours at the end of the treatment. Representative H&E images **(g)**, IHC images **(h)**, immunoblotting analysis **(i)** and TEM micrographs **(j)** of tumours that were harvested at the end of treatments. Arrows: autophagic vesicles. Scale bar in **g** and **h** is 100 μ m. Data present in **i** were from two individual tumours in each group.

General remarks:

The topic of the manuscript is of general interest in the cancer research field. All experiments were conducted properly and conclusively presented. However, the data do not support the idea of the authors of a new paradigm or platform for nanodrugs. The improvements in drug efficacy are modest and accompanied with increased toxicity. There is no convincing data on a fundamentally different profile of the nanoformulated pharmacophore compared with the Lys05 compound. Basic information such as CMC or stability of the formulation is missing. So far, no added value of the nano aspect can be seen.

Re: We thank the reviewer for the comments. In this work, we proposed an interdisciplinary drug design strategy (ONN) by incorporating the molecular self-assembly principle into the initial drug design. As a proof-of-concept, the self-delivering lysosomotropic BAQ ONNs were designed based on the principles of pharmacophore hybridization and molecular self-assembly so that they can acquire the unique advantages from the perspectives of drug discovery and drug delivery.

Compared to the parental drugs, these new BAQ entities were demonstrated to have multiple functions in cancer cells and showed enhanced anticancer activity *in vitro*. Since not belonging to the prodrug category, these entities represent a chemical structure innovation in the aspect of discovery of new chemical entities. In terms of the nanoformulation, the self-assembling property of BAQ entities provided us a very convenient method to prepare well-dispersed and homogeneous drug solutions. Otherwise, the water

solubility problem of the new BAQ entities can limit their applications as they could not be completely soluble in water as free base or hydrochloride salt forms (**Lines 83-86**).

To better explain the advantages of BAQ ONNs in the aspect of drug delivery in vivo, we firstly compared the toxicities in mice of two administration methods (ip and iv) of BAQ ONNs. We found mice were well-tolerated to BAQ ONNs by the iv injection, while the iv injection of Lys05 caused the acute death of mice, suggesting the nanoformulations of BAQ ONNs improved their safety (**Supplementary Fig. 7d,e** and **Supplementary Fig. 8**). Then we redesigned the animal treatment study in **Fig. 5**, in which mice were treated with BAQ ONNs via iv injection. A Lys05 nanoformulation, liposomes@lys05, which can also be administered into mice safely through iv injection, was added as a control. As the administration frequency was decreased from every two days to every three days, the free Lys05 did not showed therapeutic effect under this condition, but its nanoformulation, liposomes@Lys05, exhibited an improved effect on tumour inhibition that highlights the advantage of nanomedicine in drug delivery. Without interfering the body weight, the one-component BAQ12 NPs or BAQ13 NPs showed significant high antitumour efficacy over either free Lys05 or nanoformulated Lys05. These results clearly illuminated the enormous potential of BAQ ONNs in the nano aspect to improve the in vivo delivery of themselves.

For characterization of nanoparticles, we had the CMC data (called CAC in the manuscript) in **Fig. 2h** and stability data in **Supplementary Fig. 3 (Lines 129-135)**. We also deleted the relevant description about the new paradigm or platform in the revised manuscript.

References

1. Zhang, Z., Singh, R. & Aschner, M. Methods for the Detection of Autophagy in Mammalian Cells. *Curr. Protoc. Toxicol.* **69**, 20.12.21-20.12.26 (2016).
2. Wang, H., Feng, Z. & Xu, B. Bioinspired assembly of small molecules in cell milieu. *Chem. Soc. Rev.* **46**, 2421-2436 (2017).
3. Luo, M., *et al.* A STING-activating nanovaccine for cancer immunotherapy. *Nat. Nanotech.* **12**, 648 (2017).
4. Repnik, U., Stoka, V., Turk, V. & Turk, B. Lysosomes and lysosomal cathepsins in cell death. *BBA Proteins Proteom.* **1824**, 22-33 (2012).
5. McAfee, Q., *et al.* Autophagy inhibitor Lys05 has single-agent antitumor activity and reproduces the phenotype of a genetic autophagy deficiency. *Proc. Natl Acad. Sci. USA* **109**, 8253-8258 (2012).
6. Baxt, L.A. & Xavier, R.J. Role of Autophagy in the Maintenance of Intestinal Homeostasis. *Gastroenterology* **149**, 553-562 (2015).

Reviewers' comments:

Reviewer#1:

The authors have competently addressed the concerns raised in the previous review and there are no additional experiments needed for publication of this work.

Reviewer#2:

The authors have adequately addressed issues raised by this reviewer.

Reviewer#4: (Replacement for Reviewer#3)

The authors have answered reviewer #3's answers and issues in a highly satisfactory manner. We believe this work is now suitable for publication.

Response to Reviewers' comments:

Reviewer#1:

The authors have competently addressed the concerns raised in the previous review and there are no additional experiments needed for publication of this work.

Re: We thank the reviewer very much for the positive comment.

Reviewer#2:

The authors have adequately addressed issues raised by this reviewer.

Re: Many thanks for the reviewer's encouraging comment.

Reviewer#4: (Replacement for Reviewer#3):

The authors have answered reviewer #3's answers and issues in a highly satisfactory manner. We believe this work is now suitable for publication.

Re: We appreciate that the reviewer reviewed our manuscript generously.